# miR2119, a Novel Transcriptional Regulator, Plays a Positive Role in Woody Plant Drought Tolerance by Mediating the Degradation of the *CkBI-1* Gene Associated with Apoptosis

**DOI:** 10.3390/ijms23116306

**Published:** 2022-06-04

**Authors:** Furong Liu, Puzhi Zhang, Jiayang Li, Tianxin Zhang, Lifang Xie, Chunmei Gong

**Affiliations:** 1College of Life Sciences, Northwest A&F University, Yangling, Xianyang 712100, China; 17196028015@163.com (F.L.); wellbeingzpz@163.com (P.Z.); tianxinzhang@nwafu.edu.cn (T.Z.); xlf8163@163.com (L.X.); 2College of Horticulture, Northwest A&F University, Yangling, Xianyang 712100, China; jiayang@nwafu.edu.cn

**Keywords:** *Caragana korshinskii*, drought, CkamiR2119, *CkBI-1*

## Abstract

*Caragana korshinskii*, an important vegetation restoration species with economic and ecological benefits in the arid region of northwest China, is characterized by significant drought tolerance. However, the underlying molecular mechanisms by which miRNAs confer this trait in *C. korshinskii* are unclear. Here, we investigate the effect of CkmiR2119 on drought tolerance and identified its target gene, *CkBI-1*. A negative correlation of CkmiR2119 and *CkBI-1* in both stems and leaves in a drought gradient treatment followed by target gene validation suggest that CkmiR2119 might negatively regulate *CkBI-1*. Consistently, a decrease in the expression of the *CkBI-1* gene was observed after both transient transformation and stable transformation of CkamiR2119 in tobacco (*Nicotiana tabacum*). Moreover, the physiological analysis of CkamiR2119 and *CkBI-1* transgenic plants further indicate that CkmiR2119 can enhance the drought tolerance of *C. korshinskii* in two aspects: (i) downregulating *CkBI-1* expression to accelerate vessel maturation in stems; (ii) contributing to a higher level of *CkBI-1* in mesophyll cells to inhibit programmed cell death (PCD). This work reveals that CkmiR2119 can increase plants’ drought tolerance by downregulating the expression of *CkBI-1*, providing a theoretical basis to improve plants’ ability to withstand stress tolerance by manipulating miRNAs.

## 1. Introduction

Drought is one of the major abiotic stresses affecting the growth and development of plants [1] through prolonged shortages in the soil water supply [2], causing massive damage to the ecological environment and water resources [3,4]. To adapt to this extreme stress, over time plants evolved many mechanisms at the molecular level, which may act as a useful strategy for plants’ normal growth and development in a drought environment. Recently, researchers unraveled some mechanisms of plant tolerance through a variety of approaches, including genetic engineering, marker-assisted selection [5], and microRNAs.

MicroRNAs are a type of short single-stranded conserved noncoding RNAs (approximately 20–24 nucleotides in length). They derive from stem-loop regions of endogenous precursor transcripts [6,7] and participate in the modulation of many biological processes by regulating target mRNAs [8,9]. Many miRNAs play an important role in environmental stress tolerance [10,11]. Currently, some drought stress-regulated miRNAs are identified; for example, in *Arabidopsis thaliana*, overexpression of the miR398-resistant form of *CSD2* suppresses the downregulation of *CSD2* expression by miR398 and accumulates more mRNA in response to stress, thereby enhancing the scavenging effects of plant clearance ROS (reactive oxygen species) [12]. In addition, *miR393* is positively regulated by dehydration, high salt, low temperature, and ABA induction, and can mediate auxin expression [13]. In rice, *miR169g* expression may be regulated directly by CBF/DREBs to respond to drought [14]. In *Glycine max*, it was recently shown that *gma-miR398c* can negatively regulate soybean drought tolerance, providing a new site to screen for a drought tolerance resource using CRISPR [15]. In *Zanthoxylum bungeanum*, some newly discovered miRNAs can act as important regulators in the antioxidant system by inhibiting relevant genes’ expression [16]. However, the underlying mechanisms of miRNAs that respond to drought stress remain largely uninvestigated. Our previous studies found that many miRNAs may respond to drought stress in *C. korshinskii* [17]. Among them, *miR2119* is a novel miRNA reported only in leguminous plants [18]. A recent study shows that a reduction in *miR2119* expression in drought stress can increase the accumulation of its target gene *ADH1*, thereby being more responsive to this stimuli [19]. In our previous study, we predicted that *CkBI-1* may be a target gene in the drought stress regulation of CkmiR2119 [17].

Given their sessile lifestyles, plants must constantly accept environmental changes and rapidly adjust their metabolism and gene-expression profiles to adapt to inconstant conditions such as drought, extreme temperatures, salinity, nutrient imbalance, and toxic metals [20]. In past decades, various studies indicated that plants evolved sophisticated and effective systems for endomembrane trafficking in response to stress conditions, which is indispensable for the response and adaptation to environmental stresses [21]. Bax inhibitor-1 (BI-1) is an evolutionarily conserved endoplasmic reticulum localization protein crucial to the maintenance of endoplasmic reticulum homeostasis. It is widely recognized that BI-1 proteins mediate programmed cell death and resist ROS accumulation when facing biotic and abiotic stress [22]. Overexpression of the *BI-1* gene can improve the ability of plants to resist drought, high temperature, and other adverse stresses [23].

*C. korshinskii* is a main feed crop in China, with small and pinnate leaflets; it is widely distributed in the arid and semiarid regions of northwest China [24,25]. *C. korshinskii* is a suitable shrub for plantation/afforestation to restore deserted and degenerated land, and was widely planted on the Loess Plateau as part of the project to restore farmland and preserve the ecological environment [26,27]. Some recent studies found that *C. korshinskii* has the typical morphological characteristics and physiological mechanisms of drought-tolerant plants [28,29], which provides a theoretical basis for soil and water conservation, vegetation restoration, and ecosystem reconstruction in arid and semiarid areas.

In this study, we found that CkmiR2119 can downregulate *CkBI-1* in response to drought. To further identify the function of CkmiR2119, we cultivated transgenic tobacco that overexpressed CkmiR2119 and confirmed its drought tolerance using a leaf-disc tobacco experiment and other relevant plant physiological experiments. In summary, we report the differential expression of CkmiR2119 between the leaves and stems of plants in response to drought; this provides a new perspective for studies on plant drought tolerance.

## 2. Results

### 2.1. Protein Structure Prediction and Subcellular Localization of CkBI-1

To clarify the basic properties of CkBI-1, we first analyzed the three-dimensional structure of CkBI-1. The result of the structural prediction analysis indicated that the transmembrane region of CkBI-1 has a seven-fold transmembrane domain (Figure 1A,B). To investigate the distribution of CkBI-1 in leaves, the recombinant plasmid 35S: CkBI-1-GFP was constructed and transiently transformed into tobacco using *Agrobacterium* GV3101. The subcellular localization analysis suggested that CkBI-1 was mainly located in the endoplasmic reticulum membrane (Figure 1C–G). These results indicate that CkBI-1 has the same localization and function as other recognized BI-1 family members.

### 2.2. Different Expression Patterns of CkmiR2119 and CkBI-1 in Stems and Leaves of C. korshinskii

To verify the relationship between CkmiR2119 and *CkBI-1*, we first analyzed the expression patterns of CkmiR2119 and *CkBI-1* in *C. korshinskii* under gradient drought treatment using qPCR. The expression tendency of CkmiR2119 and *CkBI-1* is in the opposite direction in both stems and leaves. In stems, the relative expression of CkmiR2119 gradually increased by approximately 10 times with the increase in drought, whereas the expression of *CkBI-1* gradually decreased by 85% under the same condition. However, the relative expression of CkmiR2119 and *CkBI-1* in leaves showed a contrary tendency to that in stems (Figure 2). The results showed that CkmiR2119 was negatively correlated with *CkBI-1* in *C. korshinskii* under drought stress. Therefore, CkmiR2119 may be involved in the negative regulation of *CkBI-1*.

### 2.3. CkmiR2119 Can Cleave CkBI-1 and Reduce Its Expression Level

To study the function of CkmiR2119, *35S: CkBI-1-GUS* and *35S:* CkamiR2119 expression vectors were constructed. In the transient expression experiments, tobacco transformed with the empty vector *35S: GUS* was used as a positive control (Figure 3A); *35S: CkBI-1-GUS* and *35S:* CkamiR2119-*GUS* were injected alone (Figure 3B,C) and co-injected in tobacco leaves (Figure 3D). As expected, *35S: GUS* showed a blue signal in the whole tobacco leaf, and *35S:* CkamiR2119-*GUS* showed no blue signal. CkamiR2119 undergoes several cleavages during processing and maturation, resulting in the cleavage of the following GUS gene failing to display a blue signal. At the same time, the GUS staining of tobacco leaves injected with *35S: CkBI-1-GUS* alone was darker and distributed over entire leaves, while the staining degree of tobacco leaves injected with *35S:* CkamiR2119 and *35S: CkBI-1* was relatively light, indicating that CkmiR2119 can downregulate the expression of *CkBI-1*.

To obtain the cleavage site between CkamiR2119 and *CkBI-1*, we obtained the RNA of *C. korshinskii*. The cleaved *CkBI-1* was amplified using nested PCR by adding an adaptor to the 5′ end of the fragmented RNA and using random primers for reverse transcription. Its second-round nested PCR amplified band was approximately 300 bp (Figure 3E). After purification and recovery, it was linked to T vector for blue-white screening (Figure 3F,G). After the alignment of *CkBI-1* sequencing with miR2119, the result showed that the cleavage site of CkmiR2119 and *CkBI-1*-mRNA was located at the 10–11th nucleotide at the 5′ end of the CkmiR2119 complementary sequence (Figure 3H). Phylogenetic tree analysis and multiple sequence alignment showed that the target sequence of CkmiR2119 was relatively conserved in many legumes (Appendix A). In conclusion, the above experiments show that CkmiR2119 can cleave the transcript of *CkBI-1*, thereby negatively affecting the expression of *CkBI-1*, and that this mechanism is relatively conserved in legumes.

### 2.4. CkBI-1-OE Line Has the Effect of Inhibiting Cell Death

To verify whether *CkBI-1* can negatively regulate PCD, CkamiR2119, and *CkBI-1*, transgenic tobacco plants were obtained using *A. tumefaciens* GV3101-mediated callus culture (Appendix A). Three types of 25-day-old tobacco leaves were stained with trypan blue staining. The results showed that the staining of WT and transgenic empty vector tobacco were obviously darker than that of transgenic *CkBI-1*, which suggested that overexpression of *CkBI-1* can significantly reduce PCD of mesophyll cells (Figure 4).

### 2.5. CkamiR2119-OE Line Has More Developed Stem Xylems

Sections of CkamiR2119-OE line, wild-type, and transgenic empty vector tobacco were stained with safranin and observed under a microscope. The results showed that the CkamiR2119-OE line had significantly larger stem xylem areas and more stem vessels than the other two types of tobacco (Figure 5A–C). Additionally, we also calculated the ratios of the xylem staining area and the number of vessels of CkamiR2119-OE to WT tobacco in the above sections using ImageJ. The result was consistent with the previous section’s observation (Figure 5D,E); the proportion of the xylem area and the number of vessels in the stems of the CkamiR2119-OE line were significantly higher than those of the wild-type and empty vector plants. Moreover, after drought treatment, the relative expression of tobacco *BI-1* in the stems of WT and two transgenic tobacco were detected using qPCR. The result showed that the expression level of the tobacco *BI-1* gene of the CkamiR2119-OE lines was significantly lower than that of the other two types of plants (Figure 5F). Therefore, these results suggest that overexpression of CkamiR2119 significantly reduces the expression of the *BI-1* gene and promotes xylem formation in stems.

### 2.6. Drought Tolerance Analysis of CkmiR2119 Transgenic Plants under Drought Stress Treatment

Phenotypic analyses of transgenic and WT tobacco were performed under normal conditions and a follow-up drought treatment. In normal cultivation, CkamiR2119-OE lines had a slightly better growth state than that of the wild-type and the transgenic empty vector plants (Figure 6A). However, after drought treatment, the growth phenotype of CkamiR2119-OE lines, which had more new leaves and a lower withering rate, was obviously better than that of the other plants (Figure 6A).

Next, we collected the leaves of these three types of tobacco and analyzed their physiological parameters. The relative water content measurements revealed that there was no significant difference between the three types of tobacco when grown under normal conditions, whereas the relative water content of CkamiR2119-OE lines decreased much less than those of the other two types of tobacco after drought treatment (Figure 6B). In addition, the results of the relative electrical conductivity tests were similar to the detection of relative water content, which was a much lower increase in CkamiR2119-OE after drought treatment compared with the other two types of tobacco (Figure 6C). Therefore, these results suggest that overexpression of CkamiR2119 can improve the drought tolerance of plants.

## 3. Discussion

*C. korshinskii* has a strong drought tolerance and constitutes an important genetic resource. Plant miRNAs are critical post-transcriptional regulators, and can regulate the expression of their target genes in plants’ growth and development. At present, many miRNAs are proven to regulate plants’ life activities in extreme environments. [30,31]. In our previous study, high-throughput sequencing of miRNA and GO (Gene Ontology) enrichment analyses of *C. korshinskii* leaves under a natural precipitation gradient showed that the expression of CkmiR2119 decreased with an increase in drought gradient [17]. Surprisingly, in this study, we found that the expression tendency of CkmiR2119 in *C. korshinskii* stems was contrary to that in leaves after drought gradient treatment (Figure 2). Therefore, we first tried to identify the relationship between *CkBI-1* and CkmiR2119. Our results show that the expression of CkmiR2119 and *CkBI-1* is negatively correlated in both leaves and stems in *C. korshinskii*. In addition, the tobacco transient transformation experiment and the verification of restriction sites suggest that CkmiR2119 might be a negative regulator of *CkBI-1*. These findings suggest that CkmiR2119 can potentially have value in responding to drought, which is crucial for plants’ survival in extreme climates.

In higher plants and animals, the *BI-1* gene (*Bax inhibitor-1*) is an evolutionally conserved apoptotic inhibitor [32]. We found that *CkBI-1* has seven transmembrane domains, and it is mainly located in endoplasmic reticulum (Figure 1C–G), which is consistent with the basic features of other members in the BI-1 family [33,34,35]. Our results show that *CkBI-1* can be the target gene of *CkamiR2119* (Figure 3), suggesting that *BI-1* might play an important role in the drought tolerance of plants, which is consistent with other results [22,23,32]. The BI-1 protein is involved in endoplasmic reticulum stress [32] and maintains the homeostasis of the endoplasmic reticulum by binding to other functional proteins and inhibiting the cells’ PCD process. In our research, the results of trypan blue staining on leaves of *CkBI-1-OE* lines prove that *CkBI-1* can inhibit cell death (Figure 4). In the relative expression experiment of *CkamiR2119*-*OE* lines, overexpression of *CkamiR2119* can downregulate the expression of tobacco *BI-1* in stems (Figure 5F). Furthermore, the observation of Safranin O sections suggests that *CkamiR2119*-*OE* lines tobacco have a better formation of vascular bundles than other types of plants (Figure 5A–C).

Water transport in plants is a unified dynamic continuous system of mutual feedback: a soil–plant–atmosphere continuum (SAPC). Plants can absorb water through their roots and transport water to leaves and other organs for growth and development through sophisticated xylem systems. Extra water drafts out from the stomata and participates in the turbulent exchange in the atmosphere. As the main water conduction tissue in plants, the water transport capacity of xylem is mainly determined by the number, diameter, and length of vessels. Normally, wide vessels have a more efficient ability to transport water [36]. The formation of vessel molecules is a typical PCD process in plant growth and development [37,38,39]. Therefore, we hypothesized that overexpression of CkmiR2119 can downregulate the expression of *CkBI-1* in stems, so that it can induce more PCD and contribute to the formation of vascular bundles in plants [40], which will boost the water retention capacity, thus enhancing plants’ drought tolerance. As expected, the tobacco *CkamiR2119*-*OE* lines exhibited higher adaptation to drought stress, with significantly better detection of phenotypic and physiological indexes under drought stress. Transgenic plants had a higher relative water content in their leaves and lower conductivity than vector transgenic plants and WT plants (Figure 6).

In summary, this work reports the drought-resistant functions of CkmiR2119 in *C. korshinskii*. The results indicate that CkmiR2119 can repress the expression of *CkBI-1*, which may contribute to the drought tolerance of both stems and leaves. Based on our findings, we provide a hypothesis to explain the different expression patterns of CkmiR2119 in stems and leaves. CkmiR2119 can downregulate the expression level of *CkBI-1* by targeting *CkBI-1*-mRNA and reducing the inhibitory effect of CkBI-1 on PCD, which promotes the formation of xylem vessel molecules and increases the efficiency of water transport in stems, thereby enhancing drought tolerance. In addition, in the leaves of *C. korshinskii*, the lower expression pattern of CkmiR2119 reduces the inhibition on the higher expression of *CkBI-1*, which can inhibit the occurrence of PCD in mesophyll cells under drought stress (Figure 7). This work elucidates an important function of plant miRNAs in drought stress, and provides more evidence regarding the theoretical basis for the disclosure of drought tolerance mechanisms for improved use in arid areas to reduce the phenomenon of land desertification.

## 4. Materials and Methods

### 4.1. Plant Materials and Growth Conditions

Seeds of *C. korshinskii* were obtained from the Loess Plateau in Shaanxi Province and Inner Mongolia, Northwest China [41]. Sterilized seeds were germinated in the dark at 28 °C and were cultured in moist soil.

All the *C. korshinskii* were grown in a greenhouse with a cycle of 16 h light at 25 ± 2 °C and 8 h dark at 18 ± 2 °C. We chose 50-day-old seedlings with the same growth status and treated them with different extents of drought in different basins for 25 days; the basins were divided into three gradients, where soil water contents were maintained at 75% (suitable water availability), 55% (moderate drought), and 35% (severe drought), respectively; the soil water content (weight%) = (initial soil weight-dry soil weight)/dry soil weight × 100% = water weight/dry soil weight × 100%.

### 4.2. Protein Structure Prediction

The sequence of the CkBI-1 protein was obtained from the *C. korshinskii* transcriptome database of our laboratory. We predicted the subcellular localization of CkBI-1 protein using an online website (https://predictprotein.org/get_results?req_id=644255) (accessed on 19 August 2021). We predicted the transmembrane domain of CkBI-1 based on SOSUI (http://harrier.nagahama-i-bio.ac.jp/sosui/) (accessed on 25 March 2022). The 3D structure of CkBI-1 was predicted using SWISS-MODEL (https://swissmodel.expasy.org/interactive) (accessed on 26 March 2022).

### 4.3. Relevant Genes Cloning and Plasmids Construction and Subcellular Localization

The overexpression of CkmiR2119 was achieved by constructing artificial miRNA [42]. The miR319 precursor sequence of A. thaliana acted as the molecular skeleton of CkamiR2119. Three pairs of overlap PCR primers were designed using the online website WMD3 (http://wmd3.weigelworld.org/cgi-bin/webapp.cgi) (accessed on 25 May 2020) for sequence substitution. Total DNA was extracted from the A. thaliana seedlings using the CTAB method for cloning the AtmiR319 precursor sequence. The primers for cloning CkBI-1 were designed using Primer 5. Both CkmiR2119 and CkBI-1 genes were inserted into the vector pCAMBIA1304 between the Bgl II and Ncol I restriction sites. The relative primers are listed in Appendix A.

The constructed *pCambia1304-CkBI-1* vector was used for subcellular localization observation. At the same time, the *pCambia1304* vector was used as a control. The freeze–thaw method was used to transform relevant plasmids into *A. tumefaciens* strain GV3101. Then, we injected the bacterial liquids using 1 mL disposable syringes from the back of leaves into 4-week-old tobacco. We cultured the transgenic plants at 25 °C in the dark for one day and then had a light culture in the growth chambers at 25 °C with a 16 h photoperiod for one day. The GFP signal was observed using a confocal laser scanning microscope (Leica TCS SP8, Wetzlar, German) three days after infiltration.

### 4.4. Verification of CkBI-1 Cleaves Sites

Plant growth and RNA extraction were performed as described previously [41]. The mRNA from *C. korshinskii* leaves was purified using the PloyATtraet ^®^ mRNA Isolation system IV Kit (Promega, Madison, WI, USA). We designed RLM-5′ RACE reaction primers with reference to the FirstChoice™ RLM-RACE Kit (Invitrogen, Carlsbad, CA, USA); the 5′ junction sequence was synthesized by the Beijing Qingke Biotechnology Co., Ltd. (Beijing, China); the other primers were synthesized by the Beijing Oke Dingsheng Biotechnology Co., Ltd. (Beijing, China). The 5′ RACE connector was connected using T4 RNA ligase (Thermo Fisher Scientific, Waltham, MA, USA), and the target fragments obtained using nested PCR were connected to the pGEM-T vector (Promega, Madison, WI, USA) with the pGEM-T Easy carrier junction kit. The positive clones were screened using LB solid culture containing 40 mg/L X-Gal, 0.2 mM IPTG, and 100 mg/L Amp. We selected the positive bacteria to culture in the LB liquid medium overnight, followed by sequencing using Xi’an Optic Zesi Biotechnology Co., Ltd. (Xi’an, China).

### 4.5. Tobacco Transient Expression Experiment and Identification of Stable Expression Transgenic CkmiR2119 and CkBI-1 Tobacco Plants

All the plasmids were transformed into *A. tumefaciens* strain GV3101 using the freeze–thaw method. In the transient expression experiment, the *A. tumefaciens* solution was used to transform into 4-week-old *Nicotiana benthamiana* seedlings’ leaves with 1 mL disposable syringes. Three replicates were used for each treatment, including *pCAMBIA1304-CkBI-1*, *pCAMBIA1304*-CkamiR2119, *pCAMBIA1304*-CkamiR2119, *pCAMBIA1304-CkBI-1* cotransformation, and an empty vector infection. Then, we cultured the injected tobacco in the dark for one day followed by light culture for approximately 60 h. We clipped down the transformed leaves and soaked them in GUS staining solution for 37 °C overnight; then, we rinsed them with ethanol in gradients of 55%, 75%, and 100%, respectively.

Stable genetic transformation of CkamiR2119 and CkBI-1 genes in *Nicotiana benthamiana* by *A. tumefaciens* mediated leaf disc transformation. The sterilized tobacco leaf discs (0.5 cm × 0.5 cm) were precultivated at 25 °C for 3 d on the MS medium containing 1 mg/L 6-BA and 0.1 mg/L NAA. The pretreated leaf discs were infected with *A. tumefaciens* by soaking in transformed bacterial solution for 10 min and then moved to an MS medium containing 1 mg/L 6-BA and 0.1 mg/L NAA at 22 °C for 2 d in the dark. Then, we transferred the discs to a shoot-inducing medium (MS medium with 1 mg/L 6-BA, 0.1 mg/L NAA, 250 mg/L Crab, and 25 mg/L Hyg) for shoot differentiation, and then transferred them to a root-inducing medium (1/2MS medium with 0.1 mg/L NAA, 250 mg/L Crab, and 25 mg/L Hyg). The seedlings were planted in soil when they were 5–10 cm, and we identified the transgenic plants by genomic sequencing thereafter.

### 4.6. Quantitative Real-Time PCR (qRT-PCR) Assay

The samples of stems and leaves (approximately 0.1 g) were ground into a fine powder in liquid nitrogen and transferred to a precooled 1.5 mL EP tube. Total RNA was extracted from the seedlings using the TRIzol method [43]. The quantity and integrity of the total RNA were determined by a NanoDrop 2000 spectrophotometer (Thermo Fisher Scientific, Shanghai, China) and agarose gel electrophoresis, respectively. The cDNA sequences were obtained by a stem-loop primer using a cDNA reverse transcription kit (Takara, Tokyo, Japan). Quantitative analysis on the expression of relevant genes was performed using the TB Green^®^ Premix Ex Taq™ II (Tli RNaseH Plus) (Takara, Tokyo, Japan) and the quantitative PCR amplifier CFX96 (Biorad, Hercules, CA, USA). The qRT-PCR reaction conditions were as follows: pre-denaturation at 95 °C for 1 min, denaturation at 95 °C for 5 s, annealing and extension at 60 °C for 30 s, and amplification for 40 cycles. In this experiment, u6 and CkUB were used to design primers as the internal reference genes of miR2119 and CkBI-1, respectively. The 2^−ΔΔCT^ method and GraphPad Prism 5 software were used to calculate the relative expression of the target gene and mapping, respectively. Quantitative primers are listed in Appendix A.

### 4.7. Physiological Analysis of the Transgenic and WT Plants under Drought

All transgenic and WT seeds were cultivated in chambers under the conditions described above. Drought treatment was applied after 30 days of cultivation.

The relative water content (RWC) was calculated as (M1−M3)/(M2−M3) × 100%, where M1 = fresh weight of tobacco leaves, M2 = turgid weight (we soaked the blades in distilled water for 12 h, dried the surface of the blades with filter paper, and weighed them), and M3 = dry weight (we dried M2 in an 80 °C oven for 48 h to a constant weight). We calculated the relative water content of the leaves.

Washed fresh tobacco leaves were cut into 0.5 cm pieces and placed in a test tube with 6 mL of deionized water. All the samples were extracted with a vacuum pump and immersed. After 3 h, we measured the conductivity (E1) of the solution using a conductivity meter. After boiling the samples for 30 min, we measured the conductivity of the solution (E2) when it cooled to room temperature. We calculated the relative electrical conductivity (EC) of leaves (relative electrical conductivity = E1/E2 × 100%).

### 4.8. Safranin Staining

Transgenic tobacco stem samples were stained with Safranin O solution for 5 min after manual sectioning. After repeated decolorization with 95% ethanol, the result was observed under a light microscope.

### 4.9. Trypan Blue Staining

Leaf samples from 20-day-old tobacco seedlings of transgenic CkBI-1 gene, transgenic vector, and the WT group were immersed in trypan blue staining and boiled for 1 min. After 15 min of standing, we washed off the floating color with distilled water. The staining condition was observed under a microscope and analyzed using ImageJ.

### 4.10. Statistical Analysis

The data are presented as the mean values ± SDs (standard deviation). All the experimental data were analyzed with GraphPad Prism 5 (GraphPad Software, San Diego, CA, USA). One-way ANOVA was used to compare the statistical difference in the mean among the plant lines under different treatments based on Duncan’s multiple range test (DMRT) at a significance level of *p* < 0.05.

## Figures and Tables

**Figure 1 ijms-23-06306-f001:**
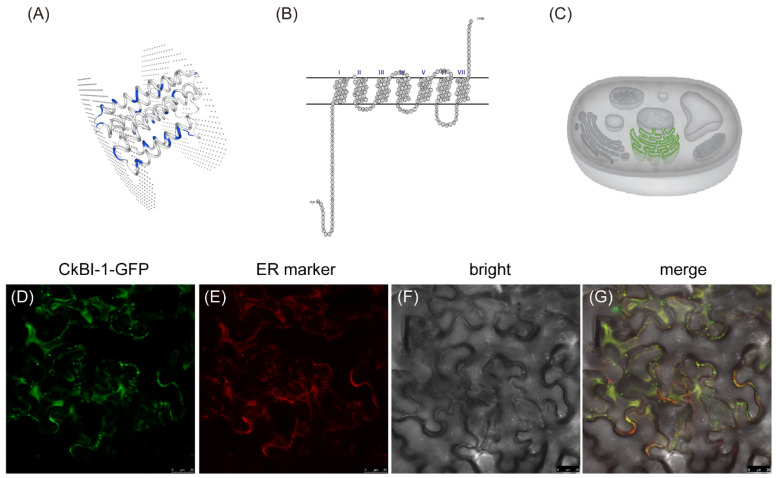
Structural analysis and subcellular localization of CkBI-1 protein. (**A**) Three-dimensional structure of CkBI-1 protein. The two atom layers represent the upper and lower surfaces of the cell membrane. (**B**) Analysis of the transmembrane domain of CkBI-1 protein. (**C**) Prediction of subcellular localization of CkBI-1 protein. (**D**–**G**) Subcellular localization of CkBI-1 protein. (**D**) GFP excitation wavelength. (**E**) mCherry excitation wavelength. (**F**) Bright field. (**G**) Displayed as an overlay of three channels. Scale bars, 25 μm.

**Figure 2 ijms-23-06306-f002:**
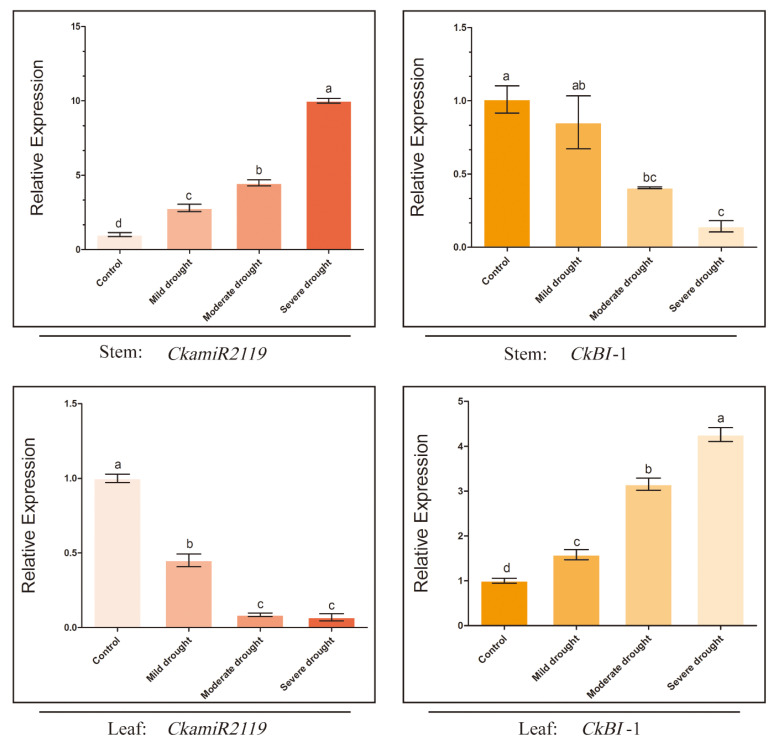
Expression trends of CkmiR2119 and *CkBI-1* in the stems and leaves of *C. korshinskii* under drought gradients. Data are shown as the mean ± SD of three independent experiments. One-way ANOVA was performed for the statistical analysis, where different letters represent significant differences (Ducan, *p* < 0.05).

**Figure 3 ijms-23-06306-f003:**
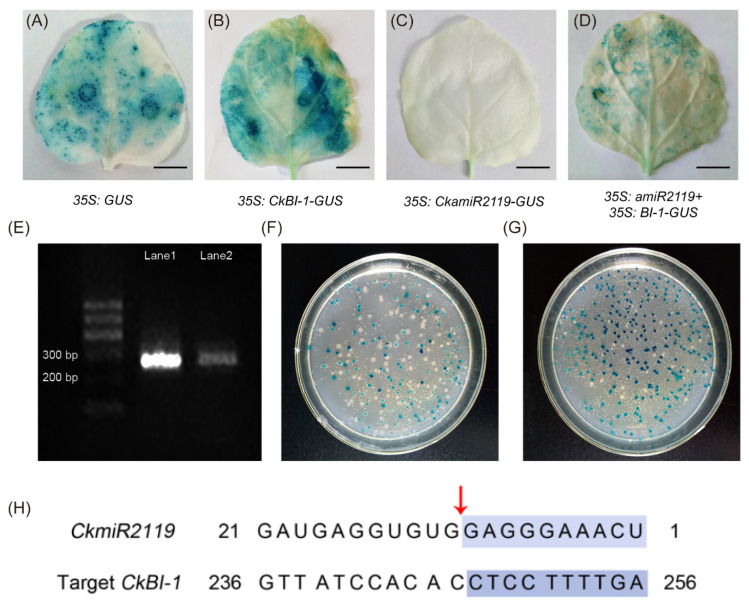
Validation of the targeting relationship between CkamiR2119 and *CkBI-1*. (**A**–**D**) Verification of the targeting relationship between CkamiR2119 and CkBI-1 transient expression in tobacco leaves. (**A**) *35S: GUS*. (**B**) *35S: CkBI-1-GUS*. (**C**) *35S:* CkamiR2119-*GUS*. (**D**) *35S:* CkamiR2119 and *35S: CkBI-1-GUS* cotransformation. (**E**) RLM-5′RACE nested PCR second-round amplification results. (**F**,**G**) Screening of positive clones after the amplified product is connected to T vector. (**H**) The cleavage site of CkmiR2119. Scale bars in (**A**–**D**), 10 mm.

**Figure 4 ijms-23-06306-f004:**
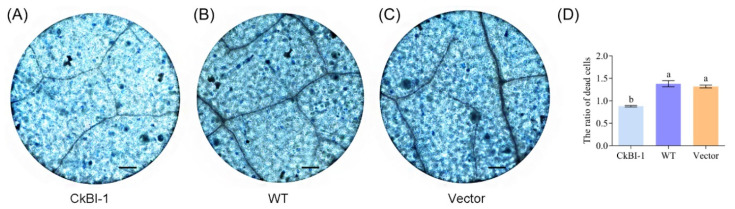
Trypan blue staining of tobacco *CkBI-1*-OE line leaves. (**A**) *CkBI-1*-OE line. (**B**) WT. (**C**) Empty vector. (**D**) Analysis of the degree of trypan blue staining. Data are shown as the mean ± SD of three independent experiments. One-way ANOVA was performed for the statistical analysis, where different letters represent significant differences (*p* < 0.05). Scale bars, 100 μm.

**Figure 5 ijms-23-06306-f005:**
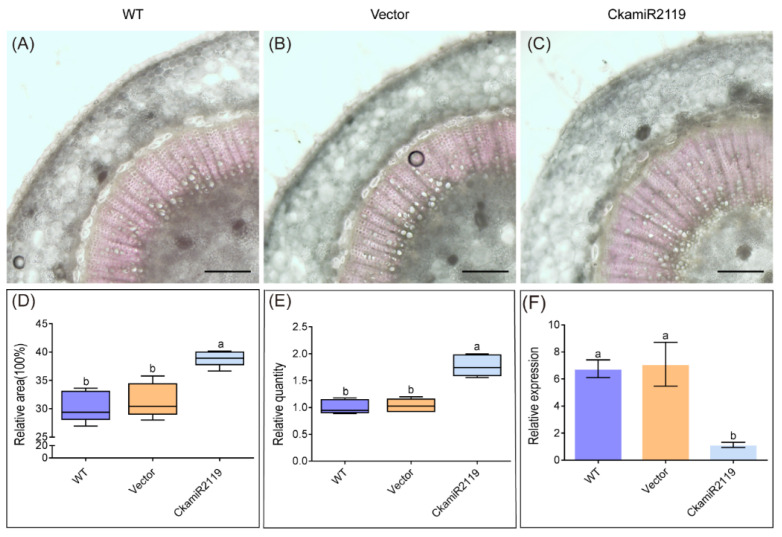
Safranin staining of freehand sections of stems of CkamiR2119-OE line under drought stress. (**A**) WT. (**B**) Empty vector. (**C**) CkamiR2119-OE line. (**D**) The relative area of the xylem in stems. (**E**) The ratio of CkamiR2119-OE line and empty vector plants to the number of vessels in the wild-type. (**F**) The relative expression of tobacco *BI-1* gene in tobacco plants after drought treatment. Data are shown as the mean ± SD of three independent experiments. One-way ANOVA was performed for the statistical analysis, where different letters represent significant differences (*p* < 0.05). Scale bars, 300 μm.

**Figure 6 ijms-23-06306-f006:**
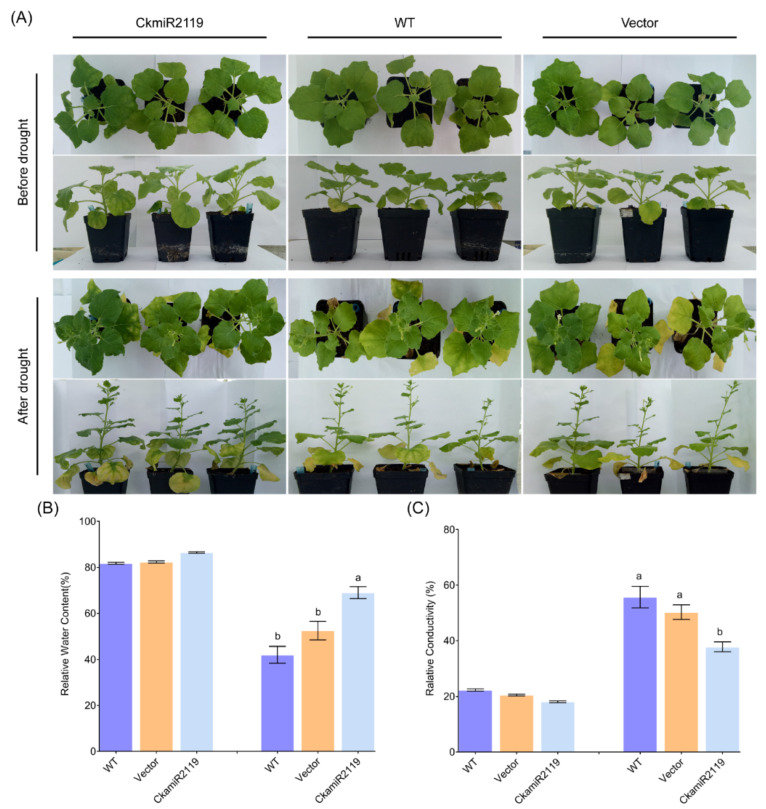
The phenotype of CkamiR2119 transgenic plants under drought treatment. (**A**) Phenotypic observations of drought treatment. (**B**) Relative water content of transgenic tobacco plants. (**C**) Relative conductivity of transgenic tobacco plants. Data are shown as the mean ± SD of three independent experiments. One-way ANOVA was performed for the statistical analysis, where different letters represent significant differences (*p* < 0.05).

**Figure 7 ijms-23-06306-f007:**
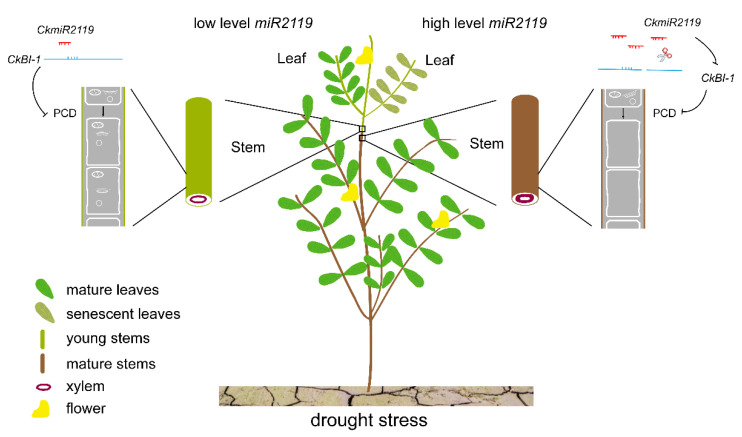
Model of CkmiR2119 regulating *CkBI-1* gene expression.

## Data Availability

The data presented in this study are available in the article or Appendix A.

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
