# Peer review of "miR2119, a Novel Transcriptional Regulator, Plays a Positive Role in Woody Plant Drought Tolerance by Mediating the Degradation of the CkBI-1 Gene Associated with Apoptosis"

_ijms, 2022, doi:10.3390/ijms23116306_

Round 1

Reviewer 1 Report

     The work by Liu et al. studies the regulation of CkBI-1 by miR2119 in Caragana korshinski and its relation to drought stress. They show a negative correlation in the expression of the miRNA and CkBI-1 gene in stem and leaves under stress. In a transient assay in tobacco, the authors show that miR2119 is capable of slicing a CkBI-1-GUS fusion mRNA. They then produced 35S:CkamiR2119 and 35S:CkBI-1 stably transformed tobacco lines. The 35S:CkBI-1 lines allowed them to show a reduction in PCD, while the 35S:CkamiR2119 lines showed an increase in stem xylems, which could be related to more PCD. Finally, the authors show that stably transformed 35S:CkamiR2119 plants display more resistance to drought.

     My main concern is related to the evidence supporting the regulatory action of CkamiR2119 on CkBI-1. This is deduced from the presence of the miRNA target sequence on the mRNA, the negative correlation between the expression of the two RNAs in C. korshinskii, in different tissues and under drought treatments of increasing severity; and from experiments in tobacco, an heterologous system. It would be important to provide more evidence obtained in C. korshinskii. For example, the 5'RACE method could be tested in C. korshinskii to detect the cleavage product of CkBI-1 under stress. It would also be interesting to bioinformatically analyze whether the miR2119 target sequence is present in orthologs of BI-1 in other legumes. The conservation of this sequence would stress the potential relevance of this regulatory mechanism.

Minor comments
     - Fig1A: the legend should explain the image in more detail. It should  mention what the two atom layers are (which I guess are sections of the outer and inner cell membrane). And it should mention that only the transmembrane region of the protein is being displayed, excluding the N and C termini (as mentioned in the main text and in Methods).
     - Fig 1B: please indicate which side is the ER lumen and which the cytosol (from literature?).
     - Fig1D,E,G: which was the purpose of exciting with mCherry wavelength if BI-1 was tagged with GFP. Please clarify.
     - Fig2: the post-hoc test applied to get the letter groupings should be mentioned.
     - Section 2.3: the word sPlice is mistaken with slice, the correct name of the cutting process directed by miRNAs. Please correct through-out (also splicesome is incorrect).
     - Fig3C: the name of the construction is 35S:CkamiR2119-GUS, I guess this is a mistake since GUS is not observed and should not be cloned here (the name of the construction mentioned in section 2.3 is correct).
     - Fig4: should be moved to supplementary material and Section 2.4 and 2.5 should be a single section.
     - Section 2.5: it is concluded that CkBI-1 reduced PCD under drought stress, but no stress to tobacco leaves is mentioned. Please clarify.
     - Methods L292-3: which tool from NCBI was used? By "functional areas" the authors mean functional regions/domains/motifs/parts of the protein? In my opinion, "area" is not the word generally used in this context.
     - Methods L297: blank control is not the best expression here, maybe "mock control" (or simply "control") or "control empty vector".

Reviewer 2 Report

The manuscript “ijms-1704502” entitled “miR2119, a novel transcriptional regulators, plays a positive role in woody plant drought tolerance by targeting degradation of CkBI-1 gene associated with apoptosis” by Liu et al. deals with an interesting subject, where the authors investigated the drought tolerance of miR2119 in Caragana korshinskii, (CkmiR2119) and identified its target gene CkBI-1 by leaf-disc tobacco (Nicotiana tabacum) experiment. The results revealed a negative correlation between CkmiR2119 and CkBI-1 in both stems and leaves in drought gradient treatment followed by the target gene validation suggested that CkmiR2119 might negatively regulate CkBI-1. Consistently, decreased expression of CkBI-1 gene was observed in both transient transformation and stable transformation of CkamiR2119 in tobacco. Moreover, physiological analysis of CkamiR2119 and CkBI-1 transgenic plants further indicated that miR2119 could enhance the drought resistance of C. korshinskii in two aspects, a) down-regulating CkBI-1 expression to accelerate vessel mature in stems, b) contributing CkBI-1 with high level in mesophyll cells to inhibit 25 programmed cell death (PCD).

For publication in the “IJMS” journal, the topic and content are appropriate. The subject of the study is interesting and topical, with high scientific and practical importance. The introduction is in accordance with the subject and correctly presented. Numerous scientific articles of recent date and in concordance to the topic of the study were consulted. The methodology of the study was clearly presented, and appropriate to the proposed objectives. The obtained results have been analyzed and interpreted in accordance with the current methodology. The discussions are appropriate, in the context of the results, and were conducted compared to other studies in the field. The scientific literature, to which the reporting was made, is recent and representative in the field. However, the review of the article revealed some minor issues, which were noted in the article and listed below:

  • Please revise the title of your manuscript to: ““miR2119, a novel transcriptional regulator, plays a positive role in woody plant drought tolerance by targeting degradation of CkBI-1 gene associated with apoptosis”
  • The abstract is long (234 words when the limit is about 200 words) and descriptive.
  • Line 1: The correct is “Article” instead of “Type of the Paper (Article)”
  • Lines 305-310: Please add information about the manufacturer of each kit.
  • Line 326: Please add information about the manufacturer of the Real-time PCR system (as in line 321).
  • Materials and Methods: Statistical analysis sub-section is missing. Please write this sub-section including the experimental layout they used in their study and the statistical software package used for the analysis. In addition, please refer to the statistical test used to determine the differences between the parameters at all treatments (e.g., Least Significant Difference test).

Thank you for your consideration.

Reviewer 3 Report

 The article entitled "MiR2119, a novel transcriptional regulator, plays a positive role in woody plant drought tolerance by targeting degradation of 

CkBI-1 gene associated with apoptosis" is intresting and may be consider for publication after the below revisions.

Please re-format the title like "Positive impact of MiR2119 on drought tolerance, a novel transcriptional regulator by targeting degradation of  CkBI-1 gene associated with apoptosis in woody ?

P.11 L.327- The PCR parameters were as follows: 37℃ for 15 min followed by  85℃ for 5 s and reserved at -80℃., Please re-check the conditions and temperature of PCR. I will suggest you re-phrase the sentence.

4.2 Protein structure prediction and subcellular localization??????????? Please explain . I don’t think this part is necessary. The authors can include or justify this part in the discussion sections. The study is related to the prediction of microRNA and its role in plant development. Authors may discuss more in the discussion sections.

The 2nd paragraph (4.2) and 4.3 must be merged into one and the proposed heading should be "cloning and transformation………… Authors may write every point under these headings. There are no means to write cleaves sites. It is a well known method for molecular biologists.

 Suggestions: Please write "qPCR" after  4.6. In this section, please mention the plant's name.

In 4.6., tobacco plant ? write the scientific name Nicotiana tabacum  or Nicotiana benthamiana

  1. 12 L.352 Stable genetic transformation of CkamiR2119 and CkBI-1 in tobacco was performed by 352 A. tumefaciens in leaf discs? What is the selection pressure? Hygromycin concentration????

Although the authors provided the gus PCR pics I will suggest that for molecular confirmation, the authors must provide the hptII PCR validation gel picture for selection of transformants ( not compulsory).

Kindly replace the pictures A ,B and C, Figure 4.

Fig 6 DPI must be increased.

In the introduction part, lines 83–89 should be in the conclusions.

2.2,  qRT-PCR may replace- qPCR

Among these, in stems, the relative  expression of CkmiR2119 was gradually increased with the increase of drought, whereas the expression of CkBI-1 was gradually decreased in the same condition? Please write the fold increment or decrease.

Fig. 3 E- replaced please.

2.4 Genomic DNA was extracted from 149 T1 generation transgenic tobacco leaves and detected using PCR amplification for screening positive seedlings.What is the gene name for PCR amplification?

Fig. 5 D. must be present like A, B and C. 9 (A) CkBI-1-OE line. (B) WT (C)  Empty vector.

Figure 5.  DPI must be increased.

Round 2

Reviewer 1 Report

The authors have properly addressed all my minor concerns. However, I do still find that my main concern was not addressed. I do understand that working with non-model species can be difficult and many experiments are hard to perform with them. I previously considered that a bioinformatics analysis of the target sequence of the miRNA would somehow strengthen the conclusion that CkamiR2119 actually regulates BI-1 in Caragana korshinskii and related species. This might not be the case, but high conservation, if present, points to functional relevance. This analysis was performed (Fig S1), but with a very reduced number of species, and the results show that some of them present mutations in this region that could affect miRNA-targeting. MiRNA-target complementarity in plants tends to be almost perfect, specially when cleavage and not translational inhibition takes place (for example, see https://doi.org/10.1146/annurev-arplant-050312-120043). I would strongly recommend the authors to add many more species to the target sequence alignment, preferably with an accompanying phylogenetic tree to sort species, and to show which of them have a better defined target sequence. And please extend the paring symbols ("|") showing which bases are complementary; as it is shown now it can be misinterpreted that the only important part is the pairing of the first bases (more similar to seed pairing, which occurs in animals).

As a minor comment, English has been improved, but still some correction should be made. For example:

- L 94: please capitalize Agrobacterium.
- L 119, title of section 2.3: it should be cleave (the verb), not cleavage.
- L 133: it should be cleaved, not cleavaged.
- L 162: observed BY a microscope. It would be better to use: under, using, etc.
- L 163-4: vessels than THE other two types of tobacco. THE is missing.

Author Response

Response letter

Thank you for your letter and the comments concerning our manuscript entitled “miR2119, a novel transcriptional regulator, plays a positive role in woody plant drought tolerance by targeting degradation of CkBI-1 gene associated with apoptosis”(Number: ijms-1704502). Those comments are all valuable and very helpful for revising and improving our paper serving as the important guiding significance to our researches. We have studied the comments carefully and have made corrections by which we hope to meet with approval. Revised portions are marked in red in the paper. The main corrections in the paper and the responses to the comments from editors are as follows:

  1. Response to comment: (I would strongly recommend the authors to add many more species to the target sequence alignment, preferably with an accompanying phylogenetic tree to sort species, and to show which of them have a better defined target sequence. And please extend the paring symbols ("|") showing which bases are complementary; as it is shown now it can be misinterpreted that the only important part is the pairing of the first bases (more similar to seed pairing, which occurs in animals).)

Response: Thank you for your careful review and constructive suggestions regarding our manuscript. Glycine max and Glycine soja are relatively closely related to Caragana korshinskii, so only a few species are considered in our last revision. The results of multiple sequence alignment shows that the ‘C’ base of their first cleavage site is conserved. However, the target site of CkmiR2119 in the BI-1 gene from the species above are relatively less conserved, so it may not be cleaved. According to the reviewer’s suggestion, we have added more species for the multiple sequence alignment in this revision, along with the corresponding phylogenetic tree analysis. Unlike the expected results, the BI-1 gene in Caragana korshinskii is more closely related to that in Phaseolus vulgaris and Arachis hypogaea than Glycine max and Glycine soja. This may also accounts for why CkBI-1, GmBI-1 and GsBI-1 are less conserved. Consistent with the phylogenetic tree results, multiple sequence alignment analysis find that the BI-1 sequence in Phaseolus vulgaris and Arachis hypogaea are more similar to that in Caragana korshinskii with few SNPs within the target of CkmiR2119, suggesting that the cleavage mechanism of CkBI-1 by CkmiR2119 may not be limited to Caragana korshinskii.

  1. Response to comment: (- L 94: please capitalize Agrobacterium.)

Response: Thank you for pointing out this mistake in our manuscript. We have revised this erroneous writing in the manuscript. (Line 93 on page 2).

  1. Response to comment: (- L 119, title of section 2.3: it should be cleave (the verb), not cleavage.)

Response: Thank you for your careful review. We have revised this erroneous writing in the manuscript. (Line 118 on page 4).

  1. Response to comment: (- L 133: it should be cleaved, not cleavaged.)

Response: Thanks for pointing out the error, which has now been corrected (Line 132 on page 5).

  1. Response to comment: (- L 162: observed BY a microscope. It would be better to use: under, using, etc.)

Response: Thank you for this valuable comment. We have changed ‘by’ to ‘under’ in the manuscript. (Line 162 on page 6).

  1. Response to comment: (- L 163-4: vessels than THE other two types of tobacco. THE is missing.)

Response: Thank you for your careful review and pointing out this grammatical error in our manuscript. We have revised this incorrect writing in the manuscript. (Lines 163-164 on page 6).

We appreciate the editors’ and reviewers’ warm work earnestly and hope that the corrections will meet with approval.

Once again, thank you very much for your comments and suggestions.

Yours

Sincerely

Chunmei Gong

Round 3

Reviewer 1 Report

     This new version of the manuscript includes an improved evolutionary analysis of the CKmiR2119 putative target site in various legumes. I think this is a valuable improvement because it includes more species and shows how this site might be more conserved in some of them. I do, however, have some questions about the accompanying phylogenetic tree. There is no methodological information regarding how this tree was constructed, in particular, from which sequences: the short DNA BI-1 fragments (not useful), the whole BI-1 CDS, the BI-1 protein, etc. The tree that I believe would be most useful is the one which shows the phylogenetic relationships among the BI-1 genes constructed from the whole protein sequences. This tree might not capture the known phylogenetic relationships among the species since BI-1 might belong to a multigenic family in legumes (in Arabidopsis I found 4 BI-1 genes). Please clarify this and change the tree in case it was done solely from the shown DNA fragment.

Minor comment, L164: still missing "the" in "vessels than other".
